# Failure of Passive Immunity Transfer Is Not a Risk Factor for Omphalitis in Beef Calves

**DOI:** 10.3390/vetsci10090544

**Published:** 2023-08-29

**Authors:** Florent Perrot, Aurélien Joulié, Vincent Herry, Nicolas Masset, Guillaume Lemaire, Alicia Barral, Didier Raboisson, Christophe Roy, Nicolas Herman

**Affiliations:** 1Haute Auvergne Veterinary Clinic, 22 ZAC Montplain Allauzier, 15100 Saint Flour, France; glemaireg@yahoo.fr (G.L.);; 2Mazets Veterinary Clinic, Les Mazets, 15400 Riom-es-Montagnes, France; aurelien.joulie@gmail.com (A.J.);; 3SELAS EVA, 79150 Argentonnay, France; v.herry@reseaucristal.fr (V.H.); n.masset@reseaucristal.fr (N.M.); 4CIRAD, UMR ASTRE, Montpellier, France, ASTRE, CIRAD, INRAE, University of Montpellier, Montpellier, Université de Toulouse, ENVT, 31300 Toulouse, France; didier.raboisson@envt.fr

**Keywords:** calves, suckler beef, cow–calf system, omphalitis, transfer of passive immunity

## Abstract

**Simple Summary:**

In calves, omphalitis is an infection of one or all the umbilical structures. It is the third most frequent disease in newborn calves. The objective of this longitudinal clinical trial was to assess the association between omphalitis and the failure of passive immunity transfer. Twenty-two cow–calf operations in central France were visited twice weekly from November 2020 to March 2021. Female (*n* = 463) and male (*n* = 501) beef calves were health scored twice: between 1 and 9 days old during the first visit and between 8 and 16 days old during the second visit. Omphalitis was defined as an enlarged umbilicus (greater than 20 mm) or pain response or an umbilical stump discharge or ultrasonographic abnormalities. During the first visit, a blood sample was collected for serum measurement of the total solids percentage (TS-%Brix) and total protein (TP). Three hundred and eleven calves (32.3%) developed omphalitis. The failure of passive immunity transfer was defined as serum %Brix < 8.1 or TP < 5.1 g/dL. No statistical association between the prevalence of omphalitis and the failure of passive immunity transfer was observed. In cow–calf systems, farm-level management factors (calving difficulty, hygiene of housing, and umbilical disinfection) seem to have more influence on the risk of this disease.

**Abstract:**

Omphalitis is the third most frequent disease in newborn calves after neonatal diarrhea and bovine respiratory disease (BRD), but limited data on the prevalence and risk factors are available in the literature. Failure of passive immunity transfer (FPIT) is recognized as a major risk factor for diseases and mortality in calves. However, the association between omphalitis and FPIT remains poorly described. To assess this association, 964 suckler beef calves from 22 farms were included in a longitudinal cohort study for 5 months. Each calf was examined twice (mean ages: 4.4 and 11.1 days old) to diagnose omphalitis through clinical examination and ultrasonographic evaluation (USE) if necessary. Measurements of the total solids percentage (TS-%Brix) and total protein (TP) were performed on the serum during the first visit to evaluate the calves’ passive immunity status. FPIT (fair and poor) was defined as serum %Brix < 8.1 or TP < 5.1 g/dL; among calves with omphalitis, 14% had FPIT and among calves without omphalitis 12% had FPIT. The omphalitis prevalence was 32.3% in calves without any other disease (overall prevalence of 30.9%). No statistical association between the prevalence of omphalitis and FPIT was observed. Further research is needed to identify the risk factors and promote the prevention measures for omphalitis in cow–calf systems, such as calving difficulty, hygiene of housing, and navel disinfection.

## 1. Introduction

Omphalitis is an infection, partial to full, of umbilical structures that can either be external (umbilical stump) or internal (vein, urachus, and arteries). Its diagnosis can be achieved through clinical examination (discharge observation or palpation) or ultrasonographic evaluation (USE) to improve the detection of internal infection [1,2,3]. Infection occurs through the ascending contamination of environmental nonspecific bacteria [4,5,6]. The resulting local damage (phlegmon, abscess, phlebitis, urachitis, arteritis, and hernia) [5,7,8] can evolve into systemic damage (peritonitis or bacteriemia associated with joint infection, uveitis, or meningitis) [9]. The consequences of omphalitis are multiple: higher risk of mortality [10,11], growth retardation [12], post-slaughter wastage [11], decreased welfare [9,13,14], and increased medical or surgical costs [7]. Omphalitis is the third most common health issue in newborn calves following digestive and respiratory diseases [10,15,16]. Its herd-level prevalence ranges between 5 and 34% [17,18], likely linked to differences in farms’ characteristics, careful and consistent umbilical cord care, and diagnostic methods.

Despite their high impact on calf health, risk factors and prevention measures related to omphalitis are poorly documented in dairy calf and hardly ever described in cow–calf systems. However, there are major differences between dairy and suckler farming, especially interactions between calves and their dam or between calves. In dairy systems, individual risk factors of omphalitis previously identified include the calving difficulty, body weight at birth, and the length of the umbilical cord [3]. Risk factors at farm level management include maternity pen hygiene, calf pen hygiene, and antiseptic umbilical cord care [3,17,19,20]. Moreover, the association between the transfer of passive immunity (TPI) and omphalitis occurrence is not clear, whereas TPI is consensually recognized as a key point to ensure good health in neonatal calves. In fact, a strong negative association between calf serum immunoglobulin G levels and the mortality rate or morbidity such as bovine respiratory disease (BRD) or calf diarrhea has clearly been demonstrated for decades [10]. Limited previous research described an association between low gammablobulin levels and omphalitis [21], but more recent publications do not support this association [10,22,23]. All the referenced studies were conducted in dairy herds with comparable farm level management: separation between calves and their dam as soon as possible, calves housed in individual pens, colostrum management protocol (exclusion of bad colostrum [10], and calves fed by bottle or esophageal tube feeder [10,22]). In cow–calf systems, colostrum management is very different: assessment of colostrum quality is very difficult, and calves often stay with their dam during the first days of life. Furthermore, the interaction between animals, especially licking, cannot be avoided. In such conditions, the association between TPI and omphalitis occurrence has never been studied. Therefore, the objective of this study was to define the association between TPI and the occurrence of omphalitis in beef calves.

## 2. Materials and Methods

### 2.1. Selection of Farms

A longitudinal cohort study was conducted from November 2020 to March 2021 in 22 French farms. All procedures were approved by the local ethical committee (registration n°2013-118). To be included, cow–calf farms planned at least 40 indoor calvings during a two-month winter period.

### 2.2. Clinical Examination and Diagnosis

Each calf was examined twice by one of the two trained veterinary practitioners (first and last authors) during 2 visits (first visit: V1 and second visit: V2). Clinical examinations included the evaluation of (i) the rectal temperature (digital thermometer), (ii) the feces consistency (1 to 4 score: firm, pasty, soupy, watery), (iii) the behavior (normal or depressed), (iv) the appetite (normal or decreased), and (v) any signs of systemic infection such as arthritis, hypopyon, or meningitis. The diagnosis of omphalitis was based on the clinical signs measured by the scoring system described in Table 1, associated with a recent study [18] and the University of Wisconsin calf health scoring system [24]. The cumulative score ranged from 0 to 6; calves were categorized as affected by omphalitis when the total score was greater or equal to one (Table 1). In detail, the umbilical stump was defined as the external part of the umbilicus surrounded by skin, and the umbilical cord is part of this, which dries in 1 to 8 days and peels off the skin after 2 to 3 weeks. The presence of umbilical stump discharge was checked (0 = no discharge, 1 = discharge). The local pain response was evaluated through palpation of the umbilical stump (firm squeeze) with one hand while the veterinarian observed the calf response (0 = no movement, 1 = flinch, kicking). The umbilical stump diameter was measured at the midpoint between the abdominal wall and the stump end using a caliper with one decimal precision (Dexter^®^, Lille, France). Abdominal umbilical structures (vein, urachus, and arteries) were assessed by deep palpation. To do this, the calf was examined in the left lateral position, and bimanual abdominal palpation was performed from the umbilical stump in craniodorsal and caudodorsal directions. An ultrasonographic evaluation (USE) was performed with a portable ultrasound device (Easi-scan: go^®^, IMV Imaging^®^, France) if the veterinarian found an abnormal thickness of one of the internal umbilical structures or when umbilical discharge was noted. The umbilical vein was measured at halfway between the umbilical ring and the liver (USE1). The urachus was sought at halfway between the umbilical ring and the bladder apex (USE2) and to its junction with the bladder apex (USE3). Each umbilical artery was measured on the lateral side of the bladder (USE4 and USE5). For each position, the cross-sectional diameter was recorded and compared with cutoff values based on previously published studies [3,25,26]. Calves with omphalitis diagnosed by one investigator were treated according to the farm’s therapeutic protocols; all treatments were recorded.

The exclusion criteria were missing data and a feces score greater than 1 to avoid a possible change in serum value.

### 2.3. Samples

During V1, the veterinary practitioner collected blood samples by jugular venipuncture into a 4 mL Vacutainer^®^ tube without anticoagulant (Improvacuter^®^, Improve, Guangzhou, China) using an 18-gauge needle (1.2 × 25 mm). Blood samples were transported to one of the two laboratories of the veterinarian office at refrigeration temperature (4–8 °C). Blood samples were allowed to clot and were centrifuged at 1500× *g* for 10 min at approximately 20 °C within 4 h following the blood sampling. Measurements of the total solids percentage (TS-%Brix) and total protein (TP) were performed on serum with two standard optical refractometers (Fioniavet^®^, Odense, Denmark; RHB-90, HHTEC, Heidelberg, Germany). Calibration using distilled water was performed between each set of analysis.

TPI values were classified into 4 categories according to the last consensus recommendations [15]: excellent (≥9.4% Brix; ≥6.2 g/dL TP), good (8.9–9.3% Brix; 5.8–6.1 g/dL TP), fair (8.1–8.8% Brix; 5.1–5.7 g/dL TP), and poor (<8.1% Brix; <5.1 g/dL).

### 2.4. Statistical Analysis

Descriptive statistics were used for the characteristics and farm management (birthplace, navel disinfection, and herd size) and calf characteristics (sex, breed, twins). A generalized linear mixed model (GLMM) was used to assess the association between the occurrence of omphalitis and TPI, with the farm as the random effect. If the calf had omphalitis at V1, V1 and V2, or V2 it was only counted once. Pearson’s chi-square test was used to evaluate the difference of the TPI categories between the sick and healthy calves’ groups. All statistical analyses were performed using R statistical software (R Core Team 2020, version 3.4.4).

## 3. Results

Forty out of the 1004 calves included in the study were excluded for different reasons (missing data, hemolyzed serum, death, or another concomitant disease). The calves’ age ranged from 1 to 9 days old at V1 and at V2 from 8 to 16 days old. The calves’ mean ages at V1 and V2 were 4.4 and 11.1 days old, respectively. The distribution of calves between farms and the individual characteristics are reported in Table 2. The farms varied in terms of the cow breed (Salers, Aubrac, Charolais, crossed), the herd size (40–160 cows), the birthplace housing (tied-stall, grouped straw bedding area, or calving pen), and the umbilical cord care (navel disinfection). Three hundred and eleven calves (32.3%) developed an omphalitis (Table 3). Among them, 1.3% of calves only had an internal structure involved, 5.4% had both external and internal structures involved, and 25.5% had an external infection. Thirty-six and 40% of calves had excellent TPI (Table 4). The correlation between the TP and %Brix to evaluate the TPI was excellent (r = 0.94).

The distribution among the four categories of TPI was the same in the groups of sick or healthy calves (chi-square test, %Brix *p* = 0.86; TP *p* = 0.63) (Figure 1).

No statistical association was found between the occurrence of omphalitis and any passive immunity transfer category used to define FPIT (*p* > 0.05) (Table 5).

## 4. Discussion

This study clearly demonstrates that FPIT is not a risk factor for omphalitis (external and/or internal) in beef cattle, which is in line with three precedent studies conducted in dairy cattle [10,23,27]. One of the explanations may be that umbilicus infection is caused by an ascending contamination from environmental germs immediately after birth under certain conditions and may not be influenced by humoral immunity. Individual factors (weight at birth, umbilical cord length, and sex) or management factors (housing hygiene and disinfection of navel) should be determinant in the occurrence of omphalitis [3,6,20]. Nevertheless, it is not excluded that the severity of the omphalitis and its complications (peritonitis, hematogenous spread of bacteria, and joint infection) may be associated with a FPIT, which should be further investigated in the future. In this study, no calf suffered complications from omphalitis probably because of the early diagnosis and treatment.

The main strength of this field study was the size of the study group (964 calves). Despite the high economic importance of omphalitis in beef cattle, very few studies have been conducted in cow–calf operations. The available studies on FPIT and omphalitis were conducted in dairy farms [10,22,23], including a small sample size of farms (*n* = 1, [22]; *n* = 2 [10]; *n* = 5 [23]). In Donovan’s study [10], only females were included, which could be a bias of interpretation because male have a higher risk of developing omphalitis [6,28]. Moreover, the diagnosis of omphalitis was not as objective (clinical sign(s) of inflammation of the umbilicus [22] or navel swollen or with discharge [10]) as the one used in this study.

FPIT is undoubtedly associated with a higher risk of mortality and morbidity in calves [10,29]. As described in many studies the TPI can be evaluated or quantified by measuring calf serum %Brix or TP [15,30,31,32]. Calves were sampled between 1 and 9 days old because the TP does not decrease or varies very little over this period [31]. The TPI was classified into four categories based on current dairy recommendations [15,33]. In the studied population, using thresholds described for dairy calves, the percentage of each farm’s calves in each category was close to the consensus recommendation: the FPIT was 12% using threshold of 8.1% Brix (dairy calves’ objective < 10%) and 37.5% if the threshold used was 8.8% Brix (dairy calves’ objective < 30%). These results suggest that in cow–calf systems without the same colostrum management (that is very widespread in dairy farms) the results are similar. In cow–calf systems, it is much more difficult to assess the same quality/practices of colostrum management than in dairy farms. Given that the dam is not milked, it is challenging to assess the colostrum quality. We excluded all calves (*n* = 5) with a modification of feces consistency at V1 because it is associated with dehydration and could modify the serum measurement.

In this study, the total incidence of omphalitis was 32.3% of all enrolled calves. This incidence is higher than previously reported by many investigators who documented the incidence of omphalitis in dairy calves to be between 1 and 14% [12,17,26,34]. However, this incidence was similar to that observed in three recent studies: from 20% to 34.2% [3,18,23]. Two main factors could explain this apparent discordance. The first one is the definition for omphalitis in our field study, which was much more inclusive than that in other studies [12,22,24]. For instance, the abdominal palpation with USE improves the detection of internal infections compared to external examination of umbilical stump only. The second possible explanation is that most studies published were conducted in dairy herds where calves are immediately separated from their dam and are usually housed in individual calf hutches [22,23]. It is likely that group-housing and interaction between dam and calves (suckling) are potential risk factors for omphalitis. In an Irish study, suckler beef calves had greater incidence of navel infection than dairy calves [35]. In this study only few cases of intra-abdominal omphalitis were diagnosed (5.4%) probably linked to the precocity of detection and the early initiation of treatment.

A scoring based on objective measures was used to mitigate the risk of discrepancies between the two operators. The objective measurement of the umbilical stump diameter with caliper has already been used in several studies [3,18,20] and is more accurate than clinical assessment: slightly enlarged or enlarged [24]. The threshold of 20 mm was arbitrarily chosen based on previously published research [1,3,19,20,25]. The evaluation of the pain response when the umbilical stump was palpated was used [3,19,24]. A wet cordon was not considered as a symptom of omphalitis because, at V1, calves were examined soon after birth (mean age 4.4 days old) and the drying times for the umbilical cord range from 1 to 8 days [36]. Then, the cutoff values for the ultrasound measure of umbilical internal structure were based on previously published research [1,2,25]. Abdominal palpation was preferred to systematic ultrasonographic examination because palpation is a sensitive diagnostic method for calves aged less than 15 days old [3].

The selection of farms was conducted irrespective of the prevalence of omphalitis during the last calving seasons and was influenced by the distance between farms and large animal practices. The farms’ characteristics and management practices were highly diverse among the selected farms (breeds, hygiene of housing, calving difficulty, and disinfection of umbilical cords). Therefore, the farm variable was used as a random effect in our univariate statistical analysis.

## 5. Conclusions

This study used an objective and systematic umbilical scoring system to enable early detection of omphalitis. The prevalence of omphalitis was 32.3%. Under the condition of the study (cow–calf operation in the center of France), FPIT was not identified as a risk factor for omphalitis. Veterinarians and farmers confronted with a high incidence of omphalitis should therefore focus on other risk factors such as calving difficulty, hygiene of housing, and navel disinfection.

## Figures and Tables

**Figure 1 vetsci-10-00544-f001:**
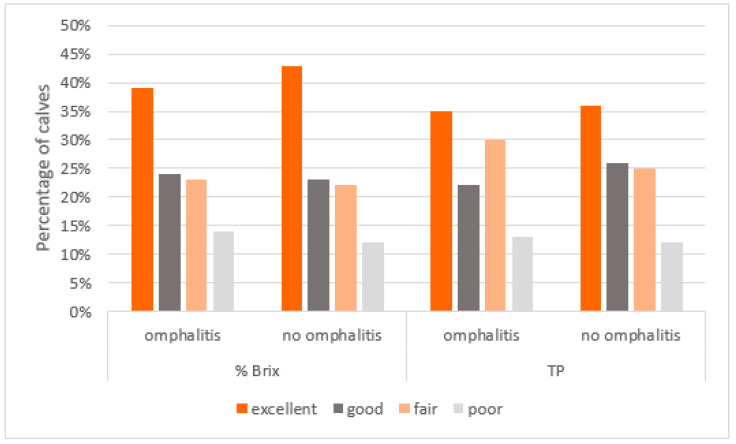
Distribution of the quality of TPI (evaluated by % Brix or TP measurement) in calves with or without omphalitis.

**Table 1 vetsci-10-00544-t001:** Clinical-signs-based scoring system.

Score	0	1
Umbilical stump diameter (mm)	<20	≥20
Pain response	No	Yes
Umbilical stump discharge	No	Yes
Ultrasonographic evaluation Diameter of vein (mm) (USE1) Urachus visualization (USE2-3) Diameter of arteries (mm) (USE4-5)	<15 (V1) < 10 (V2) No <12 (V1) < 10 (V2)	≥15 (V1) ≥ 10 (V2) Yes ≥12 (V1) ≥ 10 (V2)

USE = ultrasonography evaluation, 1–5 = position of measurements.

**Table 2 vetsci-10-00544-t002:** Descriptive statistics: characteristics of the studied population.

Item	Variable	n	%
birthplace	tied cow	178	18
	straw area	127	13
	calving pen	659	68
herd size	[40–60]	2	9
	[60–80]	3	14
	[80–100]	5	23
	[100–120]	4	18
	≥120	8	36
breed	Salers	318	33
	Aubrac	195	20
	Charolais	41	4
	crossed	410	43
sex	female	463	48
	male	501	52
twins	no	958	99
	yes	6	1
parity	primiparous	180	19
	multiparous	784	81
stump disinfection	no	767	80
	yes	197	20
antibiotic treatment	no	813	84
	yes	151	16

**Table 3 vetsci-10-00544-t003:** Descriptive statistics: characterization of omphalitis.

			N	% (/964)
V1	omphalitis		225	23.3
		external	195	20.2
		internal	6	0.6
		external and internal	24	2.5
V2	omphalitis		194	20.1
		external	162	16.8
		internal	7	0.7
		external and internal	25	2.6
V1 and/or V2	omphalitis		311	32.3
		external	246	25.5
		internal	13	1.3
		external and internal	52	5.4

**Table 4 vetsci-10-00544-t004:** Descriptive statistics: transfer of passive immunity.

		All Calves		Omphalitis		No Omphalitis	
Variable		N	Frequency %	N	Frequency %	N	Frequency %
TPI (%Brix)	excellent (≥9.4)	399	41.3	121	38.9	278	42.6
	good (8.9–9.3)	228	23.7	75	24.1	153	23.4
	fair (8.1–8.8)	214	22.2	71	22.8	143	21.9
	poor (<8.1)	123	12.8	44	14.2	79	12.1
TPI (TP g/dL)	excellent (≥6.2)	347	36.0	110	35.4	237	36.3
	good (5.8–6.1)	237	24.6	67	21.5	170	26.0
	fair (5.1–5.7)	259	26.9	93	29.9	166	25.4
	poor (<5.1)	121	12.5	41	13.2	80	12.3

**Table 5 vetsci-10-00544-t005:** Final generalized linear univariate model evaluating the TPI (with different threshold for FPIT) and the occurrence of omphalitis.

Item	Omphalitis	Coefficient	*p*-Value	OR (95% CI)
FPIT (% Brix < 7.5)	no	referent		
	yes	0.73	0.46	0.78 (0.41–1.51)
FPIT (% Brix < 7.9)	no	referent		
	yes	0.65	0.52	0.85 (0.51–1.4)
FPIT (% Brix < 8.1)	no	referent		
	yes	0.90	0.36	0.83 (0.56–1.24)
FPIT (% Brix < 8.8)	no	referent		
	yes	1.85	0.06	0.76 (0.57–1.02)

## Data Availability

The data presented in this study are available in this article.

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
