# Peer review of "Failure of Passive Immunity Transfer Is Not a Risk Factor for Omphalitis in Beef Calves"

_vetsci, 2023, doi:10.3390/vetsci10090544_

Round 1
Reviewer 1 Report
Failure of passive immunity transfer is not a risk factor for omphalitis in beef calves:
This is a well designed study, which is adequately reported. However, the manuscript needs a thorough proof reading by an English native speaker or someone with comparable English language skills.
Table 4 and figure 1 are showing the same data. One of both is sufficient.
See above
Author Response
Please find attached our manuscript reviewed by English native speaker.
Best regards
Dr Perrot Florent

Reviewer 2 Report
Thank you for this interesting study. There are some grammatical changes which need to be made plus some scientific clarification.
What is meant by "calving condition" in the introduction line 46?
Line 49/50 needs reword/edit
Multiple grammatical errors in the introduction
Line 81-83 needs rewording “In details, the umbilical stump was here defined by extra-adominal part of the umbilical structures surrounded by leather and situated between abdominal wall and the insertion of the umbilical cord”.
Line 89 can “deep palpation” be described so as to assure the reader as to its objectivity and lack of variability between operators
Line 114-115 the classification of TPI needs referencing
Line 134 “X” ?????
Line 142 this needs rephrasing to indicate statistical significance
The overall content is of interest and the number of calves in the study makes the analysis worthwhile.
The English is mostly good but there are multiple grammatical errors which need rephrasing.
Author Response
Please find attached our manuscript reviewed by an English native speaker. Furthermore all your comments and suggestions have been have been taken into account.
Best regards
Dr Perrot Florent
